



# Biases, Uncertainties, and Trends in Arctic Sea-Ice Thickness: A Cross-Product Analysis from 1995 to 2023

Valentin Ludwig[1], Caroline Ribere[2], Sara Fleury[3], Christian Haas[1], Michel Tsamados[4], Mahmoud El Hajj[2], Jerome Bouffard[5], Michele Scagliola[5], Marion Bocquet[6], Eric de Boisseson[7], Vincent Boulenger[2], Guillaume Boutin[8], Laurence Connor[9], Léo Edel[8], Stefan Hendricks[1], Ferran Hernández-Macià[10,11], Marcus Huntemann[12], Lars Kaleschke[1], Frank Kauker[1,13], Jack Landy[14], Tom Megain[3], Alek Petty[15], Till Rasmussen[16], Mads Hvid Ribergaard[16], Robert Ricker[6], Axel Schweiger[17], Hoyeon Shi[16], Xiangshan Tian-Kunze[1], Donghui Yi[18], and Alessandro Di Bella[5]

[1]Alfred Wegener Institute, Helmholtz Centre for Polar and Marine Research, Am Handelshafen 12, 27570 Bremerhaven, Germany
[2]NOVELTIS, rue du Lac 153, 31670, Labege, France
[3]Laboratoire d'Etudes en Gèophysique et Ocèanographie Spatiales, 14 avenue Edouard Belin, 31400 Toulouse, France
[4]University College London, Dept. of Earth Sciences, 5 Gower Place, London WC1E 6BS, United Kingdom
[5]European Space Research Institute, Via Galileo Galilei 2, 00044 Frascati,Italy
[6]Observing Systems, NORCE Norwegian Research Centre, Tromsø, Norway
[7]European Centre for Medium-Range Weather Forecasts, Reading, RG2 9AX, United Kingdom
[8]Nansen Environmental and Remote Sensing Center, and the Bjerknes Centre for Climate Research, Bergen, Norway.
[9]NOAA/NESDIS Center for Satellite Research and Applications, College Park, MD, USA
[10]Institute of Marine Sciences (ICM-CSIC), 08003 Barcelona, Spain
[11]isardSAT S.L., 08005 Barcelona, Spain
[12]Institute for Environmental Physics, University of Bremen, Otto-Hahn-Allee 1, 28359 Bremen, Germany
[13]O.A.Sys – Ocean Atmosphere Systems GmbH, Tewessteg 4, 20249 Hamburg, Germany
[14]Centre for Integrated Remote Sensing and Forecasting for Arctic Operations, Department of Physics and Technology, University of Tromsø The Arctic University of Norway, Tromsø, Norway
[15]Earth System Science Interdisciplinary Center, University of Maryland, MD, USA.
[16]Danish Meteorological Institute, Copenhagen, Denmark
[17]University of Washington, Applied Physics Laboratory, Polar Science Center, WA, US
[18]Global Science and Technology, LLC, 7501 Greenway Center Drive, Suite 1100, Greenbelt, Maryland 20770, USA

**Correspondence:** Valentin Ludwig (valentin.ludwig@awi.de)

**Abstract.** Sea-ice thickness is a key component of the Arctic climate system, but yet a comprehensive assessment across observations and numerical models is still missing. Previous studies have either compared only a small ensemble of sea-ice thickness products, focused on a short time scale, or both. We use an ensemble of 23 harmonised large-scale satellite, model, reanalysis and multi-product data ranging partly from 1995 to 2024, but mostly from 2010 to 2023. The products are compared against reference data derived from upward-looking sonar measurements in the Beaufort Sea. We find that biases are typically in the range of -0.2 m to 0.3 m, root mean square deviations are usually between 0.25 m and 0.5 m, and correlation coefficients mostly fall between 0.7 and 0.85, although larger deviations occur in some cases. Satellite and multi-product data mostly have lower biases and lower root mean square deviations (RMSD), but similar correlation coefficients compared to models and reanalyses. We examine the reliability of the uncertainties stated by the providers of twelve products and find that, while



individual products tend to state uncertainties that are either too small or too large relative to their actual difference towards reference data, the ensemble as a whole shows comparable magnitudes of uncertainties and difference towards reference data. Subsequently, we do a pairwise comparison between decadal averages of large-scale products. We find biases largely between 0.2 and 0.4 m, RMSDs largely between 0.4 and 0.9 m and correlation coefficients largely between 0.5 and 0.8. Our study concludes with a time-series analysis of sea-ice thickness for each category (model/reanalysis, satellite, multi-product) in November and March for 2010-2023 and 1995-2023, with the second period being limited to the region south of 81.5°N. For the first period, we find no significant trend in any category for both months. For the second period, we find that sea-ice thickness has declined by roughly 0.5-0.6 m in November and 0.3-0.4 m in March, with stronger trends for models/reanalyses and multi-product data than for satellite products.

*Copyright statement.* TEXT

# 1 Introduction

A key feature of the polar oceans is its overlying sea ice cover, which modulates energy transfer between the atmosphere and ocean. Monitoring sea ice state variability is thus key to improving our understanding of polar atmosphere-ocean interactions (Landrum and Holland, 2022; Maykut, 1986; Thielke et al., 2024), freshwater fluxes (Krishfield et al., 2014; Solomon et al., 2021) and ecosystem dynamics (Katlein et al., 2016; Sandven et al., 2025). Sea-ice thickness is relevant for, among others, biology as it governs light transmission (Light et al., 2003; Katlein et al., 2015), thermodynamics as it governs the ocean-atmosphere heat exchange (Maykut, 1986), dynamics as its local variability affects momentum transfer (Lüpkes and Gryanik, 2015) and navigation as it impedes or opens pathways for shipping (Chen et al., 2021) . It is thus a key variable for understanding and predicting the Arctic climate system (Lindsay and Zhang, 2005; Notz et al., 2016; Blanchard-Wrigglesworth et al., 2011; Notz and Community, 2020; Holland and Hunke, 2022).

The surface extent of sea ice has been monitored routinely by passive microwave radiometry with two-daily frequency since 1978 (Tonboe et al., 2016) and with daily frequency since 1987 with good accuracy (Kern et al., 2020, 2022). Measurements for sea-ice thickness, however, are less abundant. Sea-ice thickness products based on freeboard measurements (the height of sea ice above local sea level) from satellite-borne radar altimeters south of 81.5°N are available from the European Space Agency's (ESA) satellites European Remote Sensing satellite (ERS-I, 1994-1996), ERS-II (1996-2003) and Envisat (2002-2012) for latitudes until 81.5°N and since 2010 for latitudes until 88°N from the CryoSat-2 mission. Laser altimeter measurements until 88°N are provided by the Ice Cloud Elevation Satellite (ICESat, 2003-2008) and ICESat-2 (since 2018), both operated by the National Aeronautic and Space Administration (NASA). Altimetry-based products are typically aggregated over one month to achieve full Arctic-wide coverage. They derive sea-ice thickness from freeboard assuming hydrostatic equilibrium (see Data and Methods section for more details). The thickness of thin sea ice up to 1 m can be derived daily from passive microwave measurements. Passive microwave radiometers are operated on board the Soil Moisture Ocean Salinity (SMOS) and Soil





Moisture Active Passive (SMAP) satellites since 2010 and 2015, respectively. They measure the surface brightness temperature and provide sea-ice thickness up to a thickness of 1 m with high accuracy. The emissivity difference of ice and water is used to retrieve sea-ice thickness (Hernández-Macià et al., 2024; Patilea et al., 2019; Tian-Kunze et al., 2014). Model and reanalysis products range back to the 1970s, but their accuracy depends on the availability of observation data for assimilation or forcing.

Models which calculate sea-ice thickness based on solving the thermodynamic and dynamic ice growth equations are described in (Ponsoni et al., 2023; Boutin et al., 2023; Sumata et al., 2019; Sakov et al., 2012) and reanalyses are described in (Zhang and Rothrock, 2003; Schweiger et al., 2011; Zuo et al., 2019). Other products combine sea-ice thickness from models and observations (Edel et al., 2025), or from various satellite observations (Ricker et al., 2017; Shi et al., 2020).

Altimeters cover the full range of sea-ice thickness, but rely on assumptions for snow thickness and ice and snow density.
Their accuracy is higher for thick ice than for thin ice. Further, measurements need to be aggregated over one month to achieve full Arctic-wide coverage. Passive microwave radiometers provide daily temporal resolution and high accuracy for thin sea ice, but are limited to thicknesses below 1 m and have a coarse spatial resolution of typically 25 km. Models and reanalyses provide full spatial and temporal coverage, but their accuracy strongly depends on the quality of the assimilated or forcing data and the model physics. The uncertainties of raw satellite measurements, the auxiliary assumptions needed for converting
measurands to sea-ice thickness, and the different approaches for retrieving or modelling sea-ice thickness often cause large discrepancies between single products. Therefore, it is challenging to retrieve a consistent trend and magnitude from the range of methods that range from satellite data to physical and data-driven models and at last multi-product datasets. Each type of sea ice thickness estimate has their advantages and disadvantages. This manuscript compares and discusses the differences between the products with the aim of producing a better understanding of these differences.

The discrepancies between products have been assessed previously (Kwok et al., 2020; Kacimi and Kwok, 2024; Garnier et al., 2021; Sallila et al., 2019; Bunzel et al., 2018; Mallett et al., 2021) for a subset of the available products. The sea-ice thickness database from the Sea-Ice thickness products iNtercomparison eXerciSe (SIN'XS) presented by Ludwig et al. (2025) puts us in a position to assess and compare sea-ice thickness data from various sources with unprecedented comprehensiveness. We use 23 datasets with Arctic-wide coverage. They are grouped into five categories: Models and reanalyses, Ku-band altimeters,
laser altimeters, passive microwave radiometers and multi-product datasets. By multi-product, we mean datasets that combine either different satellite products or satellite and model products. They cover the time from 1994 to 2024. For evaluation, we use reference sea-ice thickness derived from three upward-looking sonars (ULS) on moorings which are deployed in the Beaufort Sea in the framework of the Beaufort Gyre Exploration Project (BGEP, Krishfield et al. (2014)).

In our contribution, we (I) evaluate the large-scale data and their uncertainties against BGEP reference data, (II) present
a pairwise comparison between large-scale products, (III) show maps of all available products for a selected month as well as decadal category-wide averages and (IV) do a time-series analysis to detect whether there has been a decline in sea-ice thickness to match the documented decline in sea-ice extent since 2007.



## 2 Data & Methods

### 2.1 Dataset Overview

We use data from the SIN'XS database (Ludwig et al., 2025). They reached out to data providers to collect standardised sea-ice thickness data. Here, we analyse the data that were collected in the SIN'XS framework. SIN'XS also contains Antarctic data, but here we focus on Arctic data. This amounts to 23 products: five models and reanalyses (HYCOM-CICE, NAOSIM, neXtSIM, ORAS5, PIOMAS), three based on passive microwave data from SMOS and SMAP, nine based on Ku-band altimetry data from ERS-I, ERS-II, Envisat and CryoSat-2, two based on laser altimetry data from ICESat and ICESat-2 and four multi-

product datasets based on combinations of data from CryoSat-2 and SMOS; from CryoSat-2, AMSR-E, AMSR2 and AVHRR; from CryoSat-2 and ICESat-2; and one from TOPAZ, CryoSat-2 and SMOS. The data are described in more detail in the next subsections. Table 1 gives an overview on the identifier used in this paper (column "Identifier"), the name of the model/sensor (column "Source"), the underlying method (column "Method"), the timespan (columns "First season" and "Last season") and a reference to the methodology for each product. All products are publicly available via: https://sinxs-tools.noveltis.fr/. :



**Table 1.** Overview on datasets used in this study. The column "Identifier" gives the label by which the dataset is referred to in the paper. The column "Source" gives the name of the sensor or model. If variables are assimilated in model runs, they are given in brackets (SST for sea-surface temperature, SIC for sea-ice concentration, T for in-situ temperature, SSS for sea-surface salinity, SLA for sea-level anomaly). Empty brackets mean that no data are assimilated. The version of the dataset is given in the column "Version". The column "Method" indicates whether the dataset is based on a model/reanalysis (Mod./Rean.), altimetry (Altim. (Ku) for Ku-band radar altimetry, Altim. (laser) for laser altimetry), passive microwave (PMW), a hybrid approach combining multiple data sources (Hybrid), or infrared satellite data (Infrared). The columns "Start" and "End" give the first and last freezing season (November to March) covered by the respective dataset. The column "Reference" provides the citation for the dataset. * The NERSC_TOPAZ4-CS2-SMOS has the name "NERSC_Topaz-Cs2-Smos" in the SIN'XS database. We have changed the naming in our paper for the sake of consistency.

| Identifier | Source | Version | Method | Start | End | Reference |
|---|---|---|---|---|---|---|
| AWI_NAOSIM | NAOSIM () | 6.3-opt5.3 | Mod./Rean. | 2002/2003 | 2021/2022 | Sumata et al. (2019) |
| DMI_HYCOM-CICE | HYCOM-CICE (SST,SIC) | 2.0 | Mod./Rean. | 1995/1996 | 2022/2023 | Ponsoni et al. (2023) |
| ECMWF_ORAS5 | ORAS5 (T,SST,SSS,SLA,SIC) | 1.0 | Mod./Rean. | 1994/1995 | 2023/2024 | Zuo et al. (2019) |
| NERSC_nextsim-opa | nextsim-opa () | 1.0 | Mod./Rean. | 1994/1995 | 2017/2018 | Boutin et al. (2023) |
| UOW_PIOMAS | PIOMAS (SIT/SIC) | 2.1 | Mod./Rean. | 1994/1995 | 2023/2024 | Zhang and Rothrock (2003) |
| CCI-C3S_CRYOSAT2 | CryoSat-2 | 3.0 | Altim. (Ku) | 2010/2011 | 2023/2024 | Hendricks et al. (2024) |
| CCI_ENVISAT | Envisat | 3.0 | Altim. (Ku) | 2002/2003 | 2011/2012 | Hendricks et al. (2018) |
| LEGOS_30YRS | ERS-1 | 2.0 | Altim. (Ku) | 1994/1995 | 1995/1996 | Bocquet et al. (2024) |
| LEGOS_30YRS | ERS-2 | 2.0 | Altim. (Ku) | 1995/1996 | 2002/2003 | Bocquet et al. (2024) |
| LEGOS_30YRS | Envisat | 2.0 | Altim. (Ku) | 2002/2003 | 2011/2012 | Bocquet et al. (2024) |
| LEGOS_30YRS | CryoSat-2 | 2.0 | Altim. (Ku) | 2010/2011 | 2022/2023 | Bocquet et al. (2024) |
| LEGOS_S3A | Sentinel-3A | 1.0 | Altim. (Ku) | 2016/2017 | 2023/2024 | Aublanc et al. (2025) |
| LEGOS_S3B | Sentinel-3B | 1.0 | Altim. (Ku) | 2018/2019 | 2023/2024 | Aublanc et al. (2025) |
| NORCE_CS2 | CryoSat-2 | 1.0 | Altim. (Ku) | 2010/2011 | 2019/2020 | Ricker et al. (2025) |
| UiT_CS2_V2.1 | CryoSat-2 | 2.1 | Altim. (Ku) | 2010/2011 | 2022/2023 | Landy et al. (2020) |
| UiT_CS2-V2.2 | CryoSat-2 | 2.2 | Altim. (Ku) | 2010/2011 | 2022/2023 | Landy et al. (2020) |
| UiT_CS2_V3.1 | CryoSat-2 | 2.3 | Altim. (Ku) | 2010/2011 | 2021/2022 | Landy et al. (2022) |
| GSFC-NSIDC_IS | ICESat | 1.0 | Altim. (laser) | 2002/2003 | 2007/2008 | Yi et al. (2011) |
| NASA_IS2 | ICESat-2 | 3.0 | Altim. (laser) | 2018/2019 | 2022/2023 | Petty et al. (2023a) |
| BEC_SMOS | SMOS | 1.0 | PMW | 2010/2011 | 2021/2022 | Hernández-Macià et al. (2024) |
| ESA_SMOS | SMOS | 3.3 | PMW | 2010/2011 | 2023/2024 | Tian-Kunze et al. (2014) |
| UB_SMOS-SMAP | SMOS-SMAP | 3.0 | PMW | 2015/2016 | 2023/2024 | Patilea et al. (2019) |
| DMI_CS2-AMSR-AVHRR | CS2-AMSR-AVHRR | 1.2 | Hybrid | 2010/2011 | 2021/2022 | Shi et al. (2024) |
| DMI_CS2-IS2 | CS2-IS2 | 1.2 | Hybrid | 2018/2019 | 2021/2022 | Shi et al. (2024) |
| ESA_CS2-SMOS | CS2-SMOS | 206 | Hybrid | 2010/2011 | 2022/2023 | European Space Agency (2023) |
| NERSC_TOPAZ4-CS2-SMOS* | TOPAZ4-CS2-SMOS | 0.1 | Hybrid | 1994/1995 | 2021/2022 | Edel et al. (2025) |



### 2.1.1 Upward-looking sonar

We use ULS data from three locations in the Beaufort Sea (75N/150W; 78N/150W;77N/140W) during all freezing seasons (October-April) from 2003/2004 until 2021/2022 as reference. The ULS measures sea-ice draft, the part of the ice below sea level. We choose ULS data because of their high native temporal sampling rate with one draft measurement per second. Draft measurements below $10\,\mathrm{cm}$ are discarded. The remaining measurements are converted to sea-ice thickness using snow thickness from the modified Warren climatology (Warren et al., 1999; Kurtz and Farrell, 2011), a daily parametrised snow density based on Mallett et al. (2020), an ice-type dependent sea-ice density based on Alexandrov et al. (2010) and a constant sea-water density of $1024\,\mathrm{kg\,m^{-3}}$. The resulting sea-ice thickness is averaged monthly and assigned to the closest pixel on a $25\,\mathrm{km}$ grid in the EASE2 projection (Brodzik et al., 2012). To assess the influence of the choice of snow thickness and sea-ice density, we re-run the calculations with varying snow thickness by $\pm\,5\,\mathrm{cm}$ and ice density by $\pm\,20\,\mathrm{kg\,m^{-3}}$ to get an envelope on the variability introduced by our choice of snow thickness and sea-ice density. A spatial representation error arises from the mismatch between pointwise reference measurements and grid data in $25\,\mathrm{km}$ grid spacing, and a temporal representation error arises from the high-frequency sampling of ULS and the monthly averages of the large-scale data. We mitigate this by averaging the large-scale data within a distance of $75\,\mathrm{km}$ around each mooring, assuming that the thickness of the ice drifting over the moorings within this area is representative of the sea-ice thickness distribution within one month. For comparison to passive-microwave only products (ESA_SMOS; UB_SMOS-SMAP; BEC_SMOS), we only use BGEP data below $1\,\mathrm{m}$.

### 2.1.2 Models and Reanalyses

There are five model and reanalysis products in our dataset: AWI_NAOSIM (Sumata et al., 2019), DMI_HYCOM-CICE (Ponsoni et al., 2023), NERSC_nextsim-opa (Boutin et al., 2023), ECMWF_ORAS5 (Zuo et al., 2019) and UOW_PIOMAS (Zhang and Rothrock, 2003). These are the models whose results are available in the SIN'XS database, but more model output would be available from the community. AWI_NAOSIM is a sea ice-ocean model that is forced using CFS reanalysis data, as well as operational data since 2011. The model runs without data assimilation, but its parameters have been optimised using a micro-genetic algorithm (Sumata et al., 2019). NERSC_nextsim-opa is an ice-ocean model forced with reanalysis data and without assimilation. PIOMAS is an ice-ocean model with a curvilinear grid design and varying resolution with an average of $40\,\mathrm{km}$. It is driven with atmospheric forcing data from the NCEP/NCAR reanalysis. Sea-ice concentration and sea-surface temperature are assimilated via a nudging/optimal interpolation procedure (Zhang and Rothrock, 2003; Schweiger et al., 2011). DMI_HYCOM-CICE is a coupled ocean and sea ice model that is forced with a mix of CARRA1 west (Schyberg et al., 2020) and ERA5. It assimilates sea ice concentration and sea surface temperatures with a slightly modified setup compared to (Ponsoni et al., 2023). ECMWF_ORAS5 is a global ocean and sea ice reanalysis that assimilates in situ temperature and salinity profiles from EN4 (Good et al., 2013), sea level anomalies from altimetry (Pujol et al., 2016) and sea surface temperature and sea ice cconcentration from OSTIA (Donlon et al., 2012). It is forced by ERA Interim between 1979 and 2015 and from ECMWF numerical weather prediction from 2015 onwards.



In AWI_NAOSIM, DMI_HYCOM-CICE, NERSC_nextsim-opa and UOW_PIOMAS, sea-ice thickness is given as grid-cell mean, while for all other products it is given as thickness of the ice which is actually present. Therefore, the model sea-ice thickness is divided by the model sea-ice concentration.

### 2.1.3 Ku-band and Laser Altimeters

Our study contains nine products based on the Ku-band altimeters ERS-I, ERS-II, Envisat and CryoSat-2, as well as two products based on the laser altimeters ICESat and ICESat-2. Altimeters measure the freeboard (the height of sea ice above local sea level), which can be converted to sea-ice thickness assuming hydrostatic equilibrium. Ku-band altimeters measure radar freeboard, while laser altimeters measure the total (sea-ice plus snow) freeboard. Retrieval algorithms for Ku-band altimetry are described in, among others, Ricker et al. (2025); Quartly et al. (2019); Paul et al. (2018); Laforge et al. (2021); Guerreiro et al. (2017); Laxon et al. (2013); Kwok and Cunningham (2008); Laxon (1994) and Shi et al. (2020). Retrieval algorithms for laser altimetry are described, for example, in Petty et al. (2023b); Yi et al. (2011); Zwally et al. (2008).

For the freeboard-to-thickness conversion, values of snow thickness, sea-ice density, snow density and sea-water density are needed, but typically poorly known. While sea-water density can be assumed to be constant, with a small uncertainty (Sievers et al., 2024), the densities of sea ice depend on the type/age of the ice, season and the degree of macro-scale porosity in deformed sea ice (Salganik et al., 2025; Jutila et al., 2022; Shi et al., 2023). Snow density depends on the composition of the snow pack (King et al., 2020) and therefore also on the season. Snow thickness can be taken from climatology (Warren et al., 1999; Kurtz and Farrell, 2011), models (Petty et al., 2018; Liston et al., 2020) or bi-frequency altimetry (Garnier et al., 2021). The Warren climatology is still widely accepted and used most often by microwave altimetry, despite its limitations (Webster et al., 2014; Lee et al., 2021).

The freeboard itself can be measured, but with significant uncertainty. For instance, radar altimeters need to determine the range elevation of a certain reference surface. The range depends mostly on the height of the backscattering targets but the angular offset from the nadir direction cannot be neglected within the freeboard uncertainty requirements. The height of the backscattered signal by Ku-Band altimeters may come from the air-snow interface, from within the snowpack, or from the snow-ice interface. Where exactly the height of mean scattering occurs depends on the composition of the snowpack (Willatt et al., 2011; Ricker et al., 2015; Guerreiro et al., 2016; Garnier et al., 2021; Shi et al., 2024) and the snow and sea-ice interface roughnesses (Landy et al., 2020). But strong, isolated off-nadir backscatter targets, such as those from young sea ice, can influence the aggregated range estimate to which the mean backscattering height is attributed. And without interferometric data to establish the off-nadir angle, significant errors in retrieved height may occur. This effect, referred to as snagging or off-nadir hooking (Quartly et al., 2019) impacts radar altimeter freeboards at both Ku and Ka frequencies. The impact of snagging depends on the satellite footprint size and thus makes it challenging to compute consistent freeboards between older pulse-limited and newer SAR altimeter types.

From the datasets used here, CCI-C3S_CRYOSAT2 and the CCI_ENVISAT (Hendricks et al., 2024) use the same snow thickness (Warren et al., 1999; Kurtz and Farrell, 2011) and densities (Alexandrov et al. (2010) for ice, Mallett et al. (2020) for snow, constant for sea water) as the ULS reference data. NORCE_CS2 (Ricker et al., 2025) uses daily along-track data



of CCI-C3S_CRYOSAT2, augmented with a drift and a growth correction and then remapped to the EASE2 projection. The 30YRS_LEGOS product (Bocquet et al., 2024) comprises a thirty-year long time series of altimeter data from ERS-I, ERS-II, Envisat and CryoSat-2 which have all been processed with the same retracker TFMRA50, the same parameters (altimetric corrections and filtering) and the same assumptions for the freeboard-to-thickness conversion parameters. The LEGOS_S3A and

155 S3B products use snow thickness from a climatology of the difference between ICESat-2 laser freeboard and Cryo-TEMPO radar freeboard and the same densities as the reference data. UiT_CS_V2.1 uses snow thickness and density from the Lagrangian Snow Evolution Model (SnowModel-LG, Liston et al. (2020)) and an ice-freeboard dependent ice density after Jutila et al. (2022). UiT_CS_V2.2 and UiT_CS_V3.1 use the same assumptions for snow thickness and densities of ice, snow and water as the reference data. The GSFC-NSIDC_IS dataset assumes constant ice ($915\,\mathrm{kg\,m^{-3}}$) and water ($1024\,\mathrm{kg\,m^{-3}}$) densi-

160 ties and a time-dependent snow density after Kwok and Cunningham (2008). NASA_IS2 uses the same ice and water densities and takes snow density and thickness from the NESOSIM model (Petty et al., 2018). The use of different assumptions for snow thickness and ice and snow densities is one cause of discrepancies between altimeter products.

### 2.1.4 Passive Microwave Radiometry

We consider three passive microwave radiometry-based products: BEC_SMOS (Hernández-Macià et al., 2024), ESA_SMOS
(Tian-Kunze et al., 2014) and UB_SMOS-SMAP (Patilea et al., 2019). BEC_SMOS uses a machine learning approach to retrieve sea-ice thickness from SMOS brightness temperatures, combined with a radiative transfer model by Burke et al. (1979). ESA_SMOS also uses SMOS brightness temperatures, with the sea-ice radiation model after Menashi et al. (1993). Both BEC_SMOS and ESA_SMOS are limited to sea-ice thickness up to $1\,\mathrm{m}$. UB_SMOS-SMAP calibrates SMAP brightness temperature measurements to SMOS sea-ice thickness after Huntemann et al. (2014), obtaining a merged product from both
sensors. All passive microwave products are originally daily retrievals which have been averaged monthly before ingestion into the SIN'XS database.

### 2.1.5 Multi-product datasets

This category comprises four products which are not based on a single satellite or model, but combine data from various sources. There are three products combining satellite data: DMI_CS2-AMSR-AVHRR, DMI_CS2-IS2 (both Shi et al. (2024))
and ESA_CS2-SMOS (Ricker et al., 2017). NERSC_TOPAZ4-CS2-SMOS (Edel et al., 2025) corrects sea-ice thickness biases of the TOPAZ4 model (Sakov et al., 2012) by assimilating the ESA_CS2-SMOS product and emulating the assimilation process prior to 2011, when ESA_CS2-SMOS data was unavailable. The daily dataset has been averaged monthly to meet the SIN'XS requirements. The DMI_CS2-AMSR-AVHRR product simultaneously estimates sea-ice and snow thicknesses by combining three satellite products, following the method of Shi et al. (2023), which has been updated to incorporate the
radar-freeboard correction described in Shi et al. (2024). The DMI_CS2-IS2 product combines CryoSat-2 and ICESat-2 to perform the simultaneous estimation using the approaches described in Shi et al. (2024). ESA_CS2-SMOS (Ricker et al., 2017) combines CryoSat-2 and SMOS data by using SMOS sea-ice thickness for ice thinner than $1\,\mathrm{m}$ and CryoSat-2 sea-ice



thickness for thicker ice via an optimal interpolation approach. Their data are originally provided as weekly composites and have been averaged to monthly resolution before ingestion into the SIN'XS database.

## 2.2 Harmonisation of datasets

We take the datasets as they are provided via https://sinxs-tools.noveltis.fr/. They are all projected to a common geographical extent using the Equal-Area Scalable Earth 2 projection (EASE2, Brodzik et al. (2012)) at 12.5 km or 25 km grid spacing. DMI_HYCOM-CICE, LEGOS_S3A, LEGOS_S3B, LEGOS_30YRS, ECMWF_ORAS5, NERSC_nextsim-opa, BEC_SMOS and ESA_SMOS are provided at 12.5 km grid spacing and downsampled by us to 25 km. All other products are provided at 25 km grid spacing. All datasets are monthly averages. For details on the data processing, we refer the reader to Ludwig et al. (2025). Furthermore, we limit our analysis to grid cells which are covered by at least 15 % sea ice. For this, we use monthly averages of the Ocean and Sea Ice Satellite Application Facility (OSI SAF) sea-ice concentration climate data record products (OSI-450-a1 from 1980 to 2020, OSI-430-a thereafter). The masks which we use in this paper are available at URL, and the OSI SAF input data are available at https://osi-saf.eumetsat.int/products/osi-430-a (OSI-430-a) and https://osi-saf.eumetsat.int/products/osi-450-a1 (OSI-450-a1).

## 2.3 Pairwise Comparison

In section 3.2, we present a pairwise comparison between datasets including all March months between 2010 and 2023. Bias, correlation and RMSD are calculated for each grid cell for each pair of products and then averaged. Different temporal and spatial coverage renders this comparison difficult: On the one hand, using only grid cells for which each dataset is available in each month of the comparison period would limit the number of samples drastically and exclude some datasets completely. On the other hand, comparing datasets regardless of temporal and spatial overlap makes it inconsistent to compare metrics between different pairs of datasets. To mitigate this, we only include pairs of datasets if they fulfil the following criteria:

1. There are overlapping measurements for at least a quarter of the ice-covered part (OSI SAF SIC > 15 %) of the Arctic Ocean

2. There are overlapping measurements in at least three out of the fourteen years

Furthermore, we do not include passive-microwave only products (ESA_SMOS; UB_SMOS-SMAP; BEC_SMOS) since they are only sensitive to sea-ice thickness up to 1 m.

## 3 Results and Discussion

### 3.1 Evaluation vs. reference data

We start our analysis by evaluating the large-scale datasets versus reference data. This is done for the satellite-only, multi-product and model/reanalysis groups. We present scatter plots for each of the groups in Fig. 1: Satellite and multi-product data



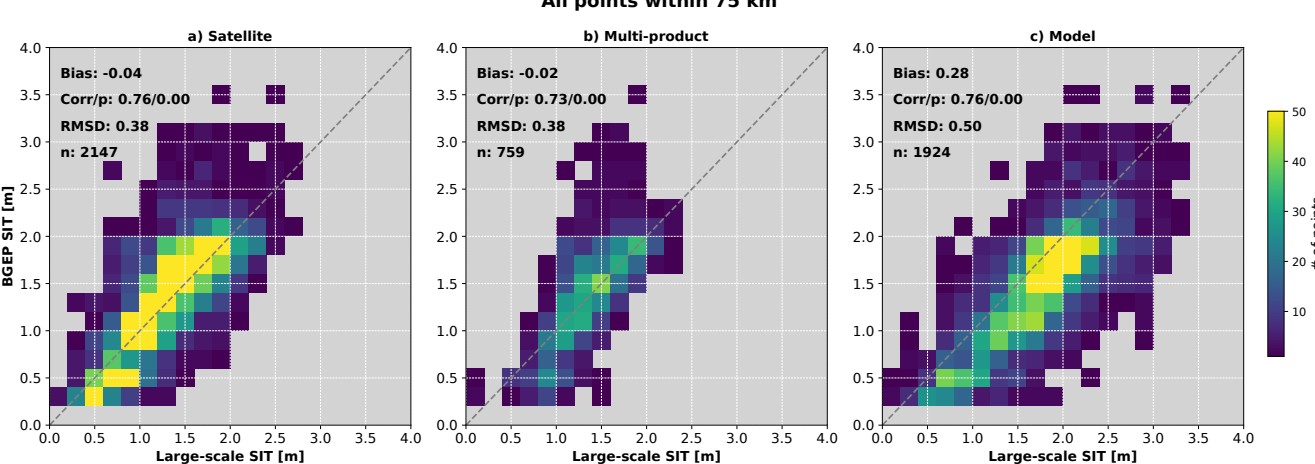

**Figure 1.** Comparison between large-scale and BGEP reference data for satellite, multi-product and model datasets (left to right). Large-scale products are shown on the x-axis, reference data on the y-axis. The coloring of bins shows how often the respective bin occurred, with the color scale given on the right. For each category, all overlapping measurements in freezing seasons (October-April) between 2002 and 2022 are shown.

show on average $0.04\,\mathrm{m}$ and $0.02\,\mathrm{m}$ thinner ice than the reference data, respectively, while the model group has a positive bias of $0.28\,\mathrm{m}$. The RMSD for the model data ($0.50\,\mathrm{m}$) is also higher than for the satellite ($0.38\,\mathrm{m}$) and multi-product ($0.38\,\mathrm{m}$) categories. The correlation coefficients of the groups are similar: 0.76 for satellite and model data, 0.73 for multi-product data.

The multi-product category hardly shows ice which is thicker than $2\,\mathrm{m}$. We attribute this to the group being mainly influenced by the ESA_CS2-SMOS product, which tends to show thinner ice than altimeter-only data by design (Ricker et al., 2017) and the NERSC_TOPAZ4-CS2-SMOS product, which corrects sea-ice thickness biases in the TOPAZ4 model (Sakov et al., 2012) prior to 2011 by combining data assimilation and machine learning based on ESA_CS2-SMOS product (Edel et al., 2025). There are two more products in the category, but with fewer data points. For a closer look into how single products perform,

we show bias, correlation and RMSD for each product in Fig. 2:

We see that most of the satellite and multi-products data have a small negative bias (underestimating the reference sea-ice thickness), while all model products overestimate the reference sea-ice thickness. Most products show correlation coefficients of 0.65 and higher, with values ranging up to more than 0.8 for some model, multi-product and satellite data. The vast majority of products shows an RMSD of less than $0.5\,\mathrm{m}$, with only two satellite products (GSFC-NSIDC_IS and CCI_ENVISAT) and

one model product (ECMWF_ORAS5) exceeding this value. The result of our comparison is impacted by the choice of the snow thickness and densities assumed for the draft-to-thickness conversion of the BGEP data. Altimeter products based on different assumptions for the freeboard-to-thickness conversion are more likely to deviate from the reference data than those that use the same assumptions.





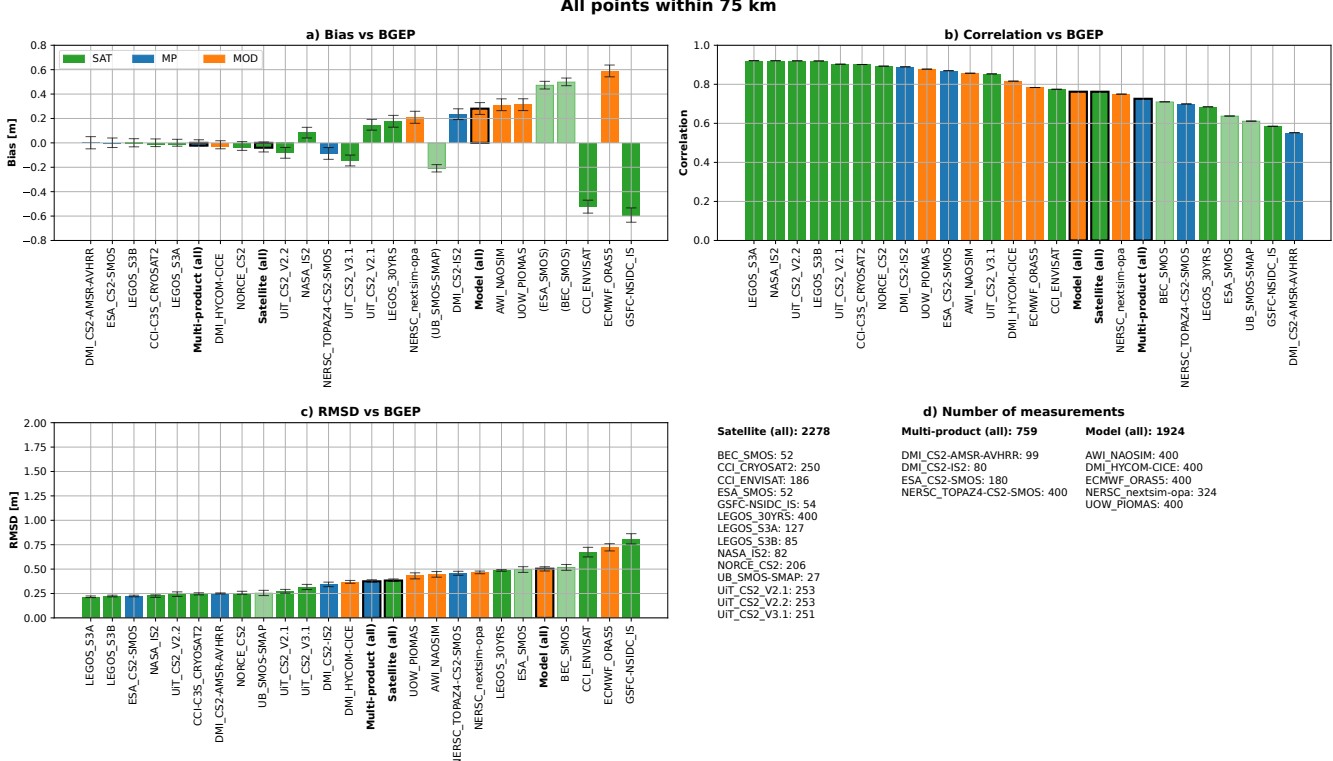

**Figure 2.** Bias, correlation and RMSD versus BGEP sea-ice thickness for each product (left to right). Biases are product-reference, i.e., a positive bias means that the respective dataset shows a higher sea-ice thickness than the reference dataset. The datasets are ordered by magnitude for each parameter separately, therefore the sequence of products is not the same. The colours give the category (see legend in a). Passive-microwave only products have not been included in Fig. 1 and are therefore shown in a paler colour and brackets around their name. The error bars on top of the bars indicate the span of the results which would be obtained with a larger reference ice thickness (obtained by decreasing snow thickness and increasing ice density) and a smaller ice thickness (obtained by increasing snow thickness and decreasing ice density).

This could be overcome by either comparing freeboard or draft instead of sea-ice thickness, or by calculating the sea-ice
thickness ourselves based on each product's freeboard where available. However, we do not have this information for all datasets. Additionally, we aim to evaluate the sea-ice thickness products as they are provided in the SIN'XS database and therefore decide to use the products as they are. To assess the influence of our choice of snow thickness and densities, we re-calculate the draft-to-thickness conversion of the BGEP data with increased and decreased snow thickness and ice density by $5\,\mathrm{cm}$ and $20\,\mathrm{kg\,m^{-3}}$, respectively. The resulting spans for bias, correlation and RMSD are indicated by the error bars in Fig.
2. We see that the bias and RMSD are influenced by our choice of snow thickness and ice density. The variability of bias and RMSD is, on average, $0.04\,\mathrm{m}$ and $0.02\,\mathrm{m}$, respectively and at most $0.06\,\mathrm{m}$ for both metrics. The correlation is not affected as perturbing the snow thickness and ice density represents a linear transformation of the draft-to-thickness conversion. Deformed





ice drifting over the moorings can lead to high ice thickness values which are not representative for the surrounding area. This can lead to outliers in the comparison between large-scale data and reference data. It could be an explanation why Fig. 1 shows
some data points with very high ice thickness in the reference data which are not matched by any large-scale product.

For a subset of products, uncertainty estimates were stated by the data providers. We evaluate these uncertainties by comparing them to the error statistics computed using the BGEP reference data in Fig. 2 to get a measure of how reliable these uncertainty estimates are. Reliable uncertainties should be on the same order of magnitude than the difference towards reference data. Averaging over all provided uncertainties, we find that the mean uncertainty (0.30 m) is close to the mean difference
(0.38 m). The spread, however, is considerable, so that the average agreement is not representative for single products. Overall, there are more pixels with too high uncertainty estimates (58 %) than pixels with too low uncertainty estimates (42 %). By too high and too low, we mean values that are larger and smaller, respectively, than the difference relative to the reference data. Looking at the products separately in Fig. 3, we see that for some products (for example 30YRS_LEGOS or CS2_NORCE) the mean uncertainty and the difference towards reference data are of similar magnitude, while for some other
products (S3A_LEGOS, S3B_LEGOS, NERSC_TOPAZ4-CS2-SMOS) there is a large discrepancy. The results are largely similar when looking at thickness classes separately (see Tab. 2).




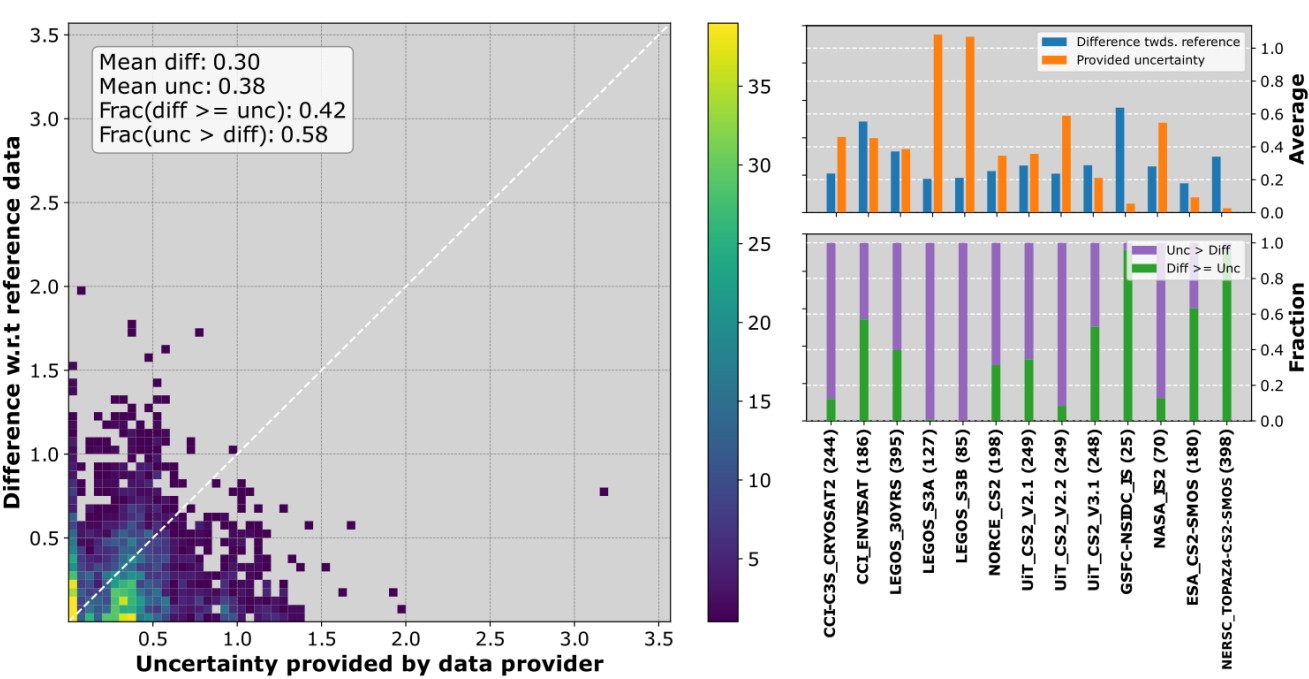

**Figure 3.** Left: Uncertainty given by the data provider versus the difference between the observation and the BGEP reference dataset. All datasets given on the x-axis of the right part of the Figure are considered together, with each grid cell for which both an uncertainty estimate and a reference measurement are available entering the scatter plot. Right, upper part: For each product separately, the average difference towards the reference data (blue) and the average uncertainty (orange). Right, lower part: For each dataset, the fraction of pixels where the uncertainty is larger than the difference towards the reference data (violet) and the fraction of pixels where the opposite is true (green). The numbers in brackets give the number of data points.





| Thickness class | Mean difference | Mean uncertainty | Fraction diff. > unc. | Fraction unc. > diff. | Number of points |
|---|---|---|---|---|---|
| 0 m-20 m | 0.30 m | 0.38 m | 0.42 m | 0.58 m | 2629 |
| 0 m-1 m | 0.22 m | 0.34 m | 0.38 m | 0.62 m | 762 |
| 1 m-2.5 m | 0.31 m | 0.4 m | 0.42 m | 0.58 m | 1807 |
| 2.5 m-20 m | 0.87 m | 0.37 m | 0.72 m | 0.28 m | 60 |

**Table 2.** Comparison of mean sea-ice thickness differences towards BGEP reference data and mean uncertainties for different thickness classes. The columns "Fraction diff. vs unc." and "Fraction unc. vs diff." give the fraction of points where the absolute difference is larger than the uncertainty, and vice versa. The last column gives the number of data points in each class.





Summarising our evaluation of large-scale datasets versus reference data, we conclude that most products have biases of less than +-0.2 m, correlation coefficients of 0.7 or higher and RMSDs of less than 0.5 m, but that there are positive and negative outliers for all metrics. The uncertainty estimates are on average on the order of the difference towards reference data, but this is not always the case for the single products.

It should be taken into account that our reference data are based on three moorings in the Beaufort Sea. The results are therefore representative for this region and the ice typically found therein, but not for other regions and ice regimes. Since no more ULS reference data are available in the SIN'XS database, we cannot extend our analysis to other regions.

## 3.2 Pairwise comparison

The next part of our study is dedicated to comparing the products among each other. We show bias, correlation and RMSD between each pair of products for all March months between 2010 and 2023 in Fig. 4, 5 and 6, respectively, with the measures described in Section 2.3 taken to ensure consistency. The metrics are computed grid point by grid point and then averaged. Fig. 4 shows that three products (DMI_CS2-AMSR-AVHRR, DMI_CS2-IS2, LEGOS_30YRS) stand out by all showing thicker ice than almost every other product, with differences between mostly 20 and 40 cm. The ESA_CS2-SMOS product shows thinner ice than all other products, but this is expected because it is more sensitive to thin ice than altimeter-only products by design. For most other products, the biases are smaller than 20 cm. Interestingly, the bias between products of different categories does not differ much from the bias between products of the same category. This underlines how sensitive altimeter retrievals are towards, for example, different assumptions for snow thickness and snow and ice density.

Correlation coefficients between products range from 0.37 (UiT_CS2_V2.1 vs DMI_HYCOM-CICE) to 0.98 (DMI_CS2-AMSR-AVHRR vs. DMI_CS2-IS2), and most of the correlation coefficients are between 0.5 and 0.8. Correlations between model and satellite products tend to be lower than those between pairs of other categories, and the highest correlations occur between multi-product and satellite data. The latter is explainable by the fact that many of the multi-product datasets assimilate satellite data, bringing them naturally close to each other. On the other hand, it is notable that even pairs of products based on the same sensor, CryoSat-2, do not necessarily have high correlation coefficients. This again points to the large role of snow thickness and snow and ice density for the freeboard-to-thickness conversion, as well as the choice of radar waveform retracker. It also highlights the necessity of dual-frequency KuKa measurements (Willatt et al., 2025), which can be used to derive snow thickness directly and will come with the launch of the Copernicus Polar Ice and Snow Topography Altimeter CRISTAL, currently scheduled for 2027. In addition, Happ et al. (2025) present a method to retrieve ice types directly from the altimeter waveform. Their method replaces the use of auxiliary products, which have previously been required for sea-ice density parametrisations in the freeboard-to-thickness conversion. Combined, these developments pave the way for a fully altimeter-based retrieval of both sea-ice thickness and snow thickness.

Akin to the correlation coefficients, RMSD values between satellite and model data (Fig. 6) tend to be higher than between products of other categories. Overall, the RMSDs span the entire range from 0.15 (DMI_CS2-AMSR-AVHRR vs DMI_CS-IS2) up to 1.02 (UiT_CS2_V2.1 vs DMI_HYCOM-CICE).





### 3.3 Comparison of large-scale datasets

Having compared the large-scale data products to reference data, we now show maps of sea-ice thickness in March averaged between 2010 and 2023, considering only products which were available in at least seven years in Fig. 7. The large-scale distribution is consistent throughout the products: thick ice up to 5 m north of the Canadian Arctic Archipelago and in the Beaufort Sea; thinner ice between 1 and 3 m in the Siberian Arctic; and the thinnest ice below 1 m in the Barents Sea. The three passive microwave-only products (Fig. 7p-r) only report sea-ice thickness up until roughly 1.2 m (BEC_SMOS, ESA_SMOS) and up until 0.5 m (UB_SMOS-SMAP) because of the limited penetration depth at 1.4 GHz and low sensitivity to thicker ice (Tian-Kunze et al., 2014; Patilea et al., 2019), in particular since the thermodynamic ice growth may exceed 50 cm in the first month and reach 1 m after three months even at moderate -20 °C (Bilello, 1961). . The median values of the other products vary between 1.27 m and 1.57 m (models), 1.34 m and 1.38 m (altimeters) and between 1.01 m and 1.83 m (multi-product). These median values likely represent the typical thickness of first-year ice. There is a larger spread, indicated by the inter-quartile range, for the model products than for the altimeters and the multi-product data. Due to the characteristic limitations to thin ice, comparing the passive microwave based products with altimeter and model data is challenging.

For one month (March 2022), we compare products averaged per category in Fig. 8. We see that the ice in the multi-year ice area north of the Canadian Arctic Archipelago is seen as thicker by the model, laser altimeter and multi-product data than by the Ku-band altimeters. In the rest of the Arctic, the magnitude is more consistent between categories. It is notable, though not surprising, that the model sea-ice thickness is much smoother than that of the other categories. The region of highest disagreement, expressed by the standard deviation maps in Fig. 8, for all groups is the Beaufort Sea, where the conditions are complicated by variable MYI advection along the northern coast of the Canadian Arctic Archipelago (Babb et al., 2022). The mean sea-ice thickness from model and laser altimeter data in this region is higher than that from the Ku-band altimeters and multi-product datasets. The standard deviation for the multi-product data is smallest, which is partly explainable because, for example, the ESA_CS2-SMOS and the NERSC_TOPAZ4-CS2-SMOS data are close to each other by design.

The patterns of differences of Ku-band altimeter data and laser altimeter data towards the category-wide mean are almost inverted: The Ku-band data show thinner ice over the multi-year ice regions and thicker ice in the seasonally ice-covered regions, while the laser altimeter data show thicker ice over the multi-year ice regions and thinner ice in the seasonally ice-covered regions. The inverted pattern may arise from the sensitivity of the estimation method to the input snow thickness (Kim et al., 2020). For Ku-band altimeters, the ice thickness increases with snow thickness used for the freeboard-to-thickness conversion, while the ice thickness decreases for the laser altimeters. A systematically too high snow thickness over multi-year ice and too low snow thickness over first-year ice would therefore lead to the observed pattern. Indeed, Lee et al. (2021) find that snow thickness increases over multi-year ice and decreases elsewhere. The thicker-than-average ice of the models in the Beaufort Sea was also reported by Dupont et al. (2015). It also manifests in the high bias from the model group in Fig. 1 and Fig. 2.





### 3.4 Time series and trend analyses

To check for trends, we look into time series of category-wide sea-ice thickness for two periods: 2010-2023, the period with most products available owing to ESA's CryoSat-2 mission in Fig. 9 and 1995-2023 in Fig. 10, the longest period spanned by

altimeters. In Fig. 9, we see that trends are generally slightly negative over time; however, there is no significant linear trend between 2010 and 2023 in any of the categories, neither at the beginning of the freezing season in November nor at the end of the freezing season in March. Correlation coefficients (i.e., how much of the variability can be explained by a linear trend) are at most 0.48, but none of them is significant at $p<0.05$. The median sea-ice thickness differs between categories both for November and March.

The picture changes when extending the time series back to 1995 (Fig. 10), the starting time of the 30-year time series by Bocquet et al. (2024). Before this, only model data would be available. By extending the time series backwards, we see the sea-ice thickness declining at a rate of -0.02 m/yr (Ku-band altimeters and models) and at a rate of -0.03 m/yr (multi-product data) in November. The slopes in March are less for Ku-band altimeters (-0.01 m/yr) and similar for multi-product data. All trends are significant, with correlation coefficients between 0.51 and 0.74 in November and between 0.28 and 0.76 in

March. We see a narrower spread of the altimeter data (smaller inter-quartile range in the boxplots) between 2000 and 2010 than afterwards, but this is probably due to the number of products being included: Before 2002, the only altimeter product is the LEGOS_30YRS product by Bocquet et al. (2024), between 2002 and 2010 there is also the CCI_ENVISAT product and after 2010 the CryoSat-2 products are also available . Additionally, the LEGOS_30YRS product only goes up to 81.5°N for the entire period, while other CryoSat-2 products go up to 88°N. The trend in Fig. 10 does not seem to be linear over time,

rather there is a stronger decrease until 2007 and a levelling off afterwards. This is in line with other studies (Sumata et al., 2023; Smedsrud et al., 2017). Based on the full ensemble, the Arctic sea ice cover has thinned by approximately 50-60 cm in November and 30-40 cm in March between the 1990s and 2020s.

We also examined whether the fact that there is apparently no significant trend in sea-ice thickness since 2010 is simply due to the time series being too short. To see how the significance of the slope changes over time, and if it is at all possible

to detect significant trends in a 13-year time series, we did a linear regression over each 13-year period since 1980, starting with 1980-1993 and ending with 2010-2023. Fig. 11 shows that there were significant trends also over this shorter timescale for starting years between 1985 and 2000 for all categories in November and March, with only a few exceptions. The temporal evolution in November and March is similar, but the March trends are roughly -0.02 m/yr stronger. The slopes are consistent for time windows of 2005-2018 and later, but the trends in these periods are not significant. We conclude from this analysis

that the statement that there is no trend in sea-ice thickness between 2010 and 2023 is robust, and not just owed to the short length of the period.

A similar study has been performed by England et al. (2025) for sea-ice extent. They find that the observed pause in sea-ice extent decline in the last twenty years is statistically robust. Stern (2025) reports that there was a regime change in sea ice extent in 2007, with no significant trend afterwards and examines several physical explanations from literature, for example

interdecadal variability masking out anthropogenically induced melting (Baxter et al., 2019) or less sea-ice export through



Fram Strait (Francis and Wu, 2020). Still, Stern (2025) notes that the younger and thinner sea ice since 2007 may lead to another regime shift towards even thinner ice, consistent with the statement of the Intergovernmental Panel on Climate Change (IPCC) (2023) that the first practically ice-free Arctic summer is *likely* to be expected before 2050 with $high confidence$.

## 4 Conclusions

We analyse an ensemble of 23 sea-ice thickness products from models, reanalyses, satellite measurements and multi-product data. The analysis comprises an evaluation of the products and their uncertainties against ULS reference data, a comparison between the products themselves and a time-series analysis. Our main findings are:

– Satellite and multi-product data show comparable statistics and outperform model products when compared to reference data. The biases are -0.02 m/-0.02 m/0.28 m, the correlation coefficients are 0.76/0.73/0.76 and the RMSD values are
0.38 m/0.38 m/0.50 m for satellite, multi-product and model data, respectively.

– The uncertainties provided with the products are on average on the same order of magnitude as the error towards reference data, but this does not necessarily hold for single products.

– Comparing decadal averages at the end of the freezing seasons between all pairs of products shows biases between 0.2 and 0.4 m, correlations between 0.4 and 0.9 and RMSDs between 0.4 and 1 m aggregated over all categories. The
bias between products is almost independent of the category, for correlation and RMSD we find the largest agreement between satellite and multi-product data, and the smallest agreement between satellite and model products.

– There is no significant linear trend in November and March sea-ice thickness between 2010 and 2023 in any category.

– There is an overall decline of 0.5 m-0.6 m in November and of 0.3-0.4 m between 1995 and 2023, with a significant linear trend in all three categories for both months.

## 5 Outlook

The purpose of this study is to evaluate the sea-ice thickness of all the products that have been integrated into the SIN'XS project database in order to determine the extent to which these different solutions agree among each other and whether they correspond to what can be observed locally. We have thus decided to use directly the sea-ice thickness provided. However, a lack of knowledge about snow thickness can lead to significant errors in ice thickness estimates and the diversity of available
snow thickness products may lead to differences beyond the quality of the freeboard measurement taken.

As there is currently no consensus on which snow thickness product to use for converting the BGEP draft data to sea-ice thickness, we used Warren's climatology, which is still the most widely used solution in the field. We know that this climatology is no longer truly relevant due to the effects of climate change.

We therefore recommend that, using this unique database already gathered, a study similar to this one is conducted on snow
thickness in order to determine the most appropriate solutions at this time. Indeed, several snow thickness products have been





already provided and most of the sea-ice thickness products also include the snow thickness used. The best snow thickness solutions can then be selected among these products to calculate sea-ice thickness and drafts from measured freeboard and to repeat the analysis on ice thickness. This would also be an opportunity to standardise the choice of ice density, the uncertainty of which has as significant an impact on the sea-ice thickness as the uncertainty of freeboard measurement. Beyond thickness,

an analysis of sea ice volume variations can provide a better understanding of trends thanks to lower sensitivity to the surface area under consideration.

The analysis also focused on the Beaufort Gyre using the BGEP moorings. We would like to extend it to other reference measurements, particularly airborne measurements (Icebird, Cryo-TEMPO, CryoVEx, OIB) and more ULS data (Lindsay, 2010) to account for a broader range of sea-ice types and regions. In particular, we would like to extend these studies to

Antarctica. One of the goals of future projects should therefore be to gather more reference measurements, particularly snow thickness and austral measurements, in the SIN'XS database.

**Appendix A: Tables**





**Figure 4.** Pairwise comparison of biases between products, considering all March months between 2010 and 2023. Pairs are only included if they are/were available in at least 3 years, and if the region covered by both products amounts to at least 25% of the ice-covered region. The bias is calculated as the difference between the product on the y-axis and the product on the x-axis, i.e., a positive bias indicates that the product on the y-axis shows thicker ice. The products are sorted by category (model, satellite, multi-product from top to bottom) and, within categories, sorted alphabetically. The categories are indicated by the black vertical/horizontal lines.





**Figure 5.** Pairwise correlation between products, considering all March months between 2010 and 2023. Design as in Fig. 4.





**Figure 6.** Pairwise RMSD between products, considering all March months between 2010 and 2023. Design as in Fig. 4.





**Figure 7.** a)-r): Average March sea-ice thickness between 2010 and 2023 per product. The years in which each product is available are given in the panel's title. a)-e) are models, f)-l) are altimeters, m)-o) are multi-product data and p)-r) are passive-microwave products. Note the different colorbar for p)-r). The boxplots in s) give the statistical distribution of each product, with the median given by the horizontal black line, the inter-quartile range (25-75%) given by the height of the boxes, the whiskers showing 1.5 times the inter-quartile range and empty circles showing values above and below. The colours indicate the category (see legend). Note that two products (LEGOS_30YRS-CS2, LEGOS_S3A) only go until 81.5°N, they are indicated by more transparent boxes and a bracket around their name. For the products labelled with an asterisk, there are outliers which exceed the y-axis. The fraction of values exceeding the y-axis (at maximum 0.39 %) and the maximum sea-ice thickness are given in Table A1 in the Appendix.



**Figure 8.** Sea-ice thickness in March 2022, averaged across categories the entire ensemble and per category, with the category given in each column's title (upper row). The second row shows the standard deviation between the products of each category. The third row shows the difference between the mean of each category and the mean across all four categories the entire ensemble. The models comprise AWI_NAOSIM, UOW_PIOMAS, DMI_HYCOM-CICE and ECMWF_ORAS5; the Ku-band altimeters comprise CCI_CRYOSAT2, LEGOS-30YRS UiT-CS2_V2.1, UiT-CS2_V2.2 and UiT-CS2_V3.1, the laser altimeter is NASA_IS2 and the multi-product data comprise DMI_CS2-AMSR-AVHRR, DMI_CS2-IS2, ESA_CS2-SMOS and NERSC_TOPAZ4-CS2-SMOS.




**Figure 9.** Boxplot of sea-ice thickness per category for November (upper part) and March (lower part), comprising all available products from the respective category and year between 2010 and 2023. All pixels in the northern hemisphere with a sea-ice concentration of at least 15 % are considered. The dashed lines show a linear regression over the means of each category, with slope, correlation coefficient and p-value given in the upper left




**Figure 10.** Boxplot of sea-ice thickness per category for November (upper part) and March (lower part), comprising all available products from the respective category and year between 1995 and 2023. All pixels in the northern hemisphere with a sea-ice concentration of at least 15 % are considered. The dashed lines show a linear regression over the means of each category, with slope, correlation coefficient and p-value given in the upper left.





# Slopes of LR for 13-year period

## November

## March

**Figure 11.** Each dot marks the slope of a linear regression over mean Arctic-wide sea-ice thickness, starting in the respective year over the next 13 years (i.e., the dot at year 1980 gives the results of a linear regression from 1980 until 1993). The colours indicate the category according to the legend. Points with reduced opacity indicate insignificant trends (p-values above 0.05). The upper panel shows November values, the lower panel shows March values.





| Dataset | Percentage above 6 m | Maximum sea-ice thickness (m) |
| --- | --- | --- |
| AWI_NAOSIM | 0.0045 % | 6.11 |
| DMI_HYCOM-CICE | 0.0169 % | 24.69 |
| NERSC_nextsim-opa | 0.0286 % | 7.69 |
| CCI_CRYOSAT2 | 0.0241 % | 7.65 |
| LEGOS_30YRS-CS2 | 0.0482 % | 12.63 |
| LEGOS_S3A | 0.3929 % | 13.65 |
| NORCE_CS2 | 0.1346 % | 9.88 |
| UiT_CS2-V2.1 | 0.0297 % | 7.96 |
| UiT_CS2-V2.2 | 0.0372 % | 8.88 |

**Table A1.** Supplement to Fig.7. Gives the percentage of values exceeding the y-axis in Figure 7s, and the corresponding maximum sea-ice thickness.



**A1**

*Author contributions.* VL: Conceptualization, Data curation, Formal analysis, Investigation, Methodology, Software, Validation, Visualiza-
tion, Writing (original draft preparation)

CR: Conceptualization, Project administration, Supervision

SF: Conceptualization, Data curation, Funding acquisition, Methodology, Supervision, Writing (review and editing)

CH: Conceptualization, Funding acquisition, Methodology, Supervision

MT: Conceptualization, Funding acquisition, Methodology, Supervision

MEH: Conceptualization, Funding acquisition, Supervision

JB: Conceptualization, Methodology, Supervision

MS: Conceptualization, Methodology, Supervision

MB, VB, LC, FHM, MH, MHR, LK, XT, DY: Data curation

SH: Conceptualization, Data curation, Funding acquisition, Methodology, Supervision, Writing (review and editing)

EdB, GB, LE, FK, JL, AP, RR, TR, AS, HS: Data curation, Writing (review and editing)

ADB: Conceptualization, Methodology, Supervision

*Competing interests.* Christian Haas, Lars Kaleschke and Michel Tsamados are members of the editorial board of The Cryosphere.

*Disclaimer.* The scientific results and conclusions, as well as any views or opinions expressed herein, are those of the authors and do not
necessarily reflect those of NOAA or the U.S. Department of Commerce.

*Acknowledgements.* This work has been funded by the ESA SIN'XS project. The project team collectively acknowledges the data provision
through the institutes given in the manuscript body. The large language model ChatGPT has been used for suggestions for the title. The ulti-
mate choice, inspired by but not copied from these suggestions, was made by the authors. The ULS sea-ice draft data were collected and made
available by the Beaufort Gyre Exploration Program based at the Woods Hole Oceanographic Institution (https://www2.whoi.edu/site/beaufortgyre/)
in collaboration with researchers from Fisheries and Oceans Canada at the Institute of Ocean Sciences. The measurements were funded by
the National Science Foundation, Office of Polar Programs. Valentin Ludwig acknowledges funding from the ESA SIN'XS project. Marion
Bocquet was supported by CNES (contract no. 3342) and CLS (contract no. CLS-ENV-BC-22-0221) thanks to a doctoral fellowship. Fer-
ran Hernández-Macià acknowledges funding from the ARCTIC-MON project (PID2021-125324OB-I00); Agencia Estatal de Investigación
(AEI), Spain. Guillaume Boutin was supported by the MuSIC project (NRC grant no 325292) and the SASIP project, supported by Schmidt
Sciences. Léo Edel was supported by the project TARDIS (grant no. 325241), funded by a Norges Forskningsråd grant. Alek Petty was sup-
ported by a NASA ICESat-2 Science Team award (grant no. 80NSSC23K1253). Robert Ricker was supported by ESA CCI Sea Ice (CCN-2



to contract 4000126449/19/I-NB -Sea_Ice_cci). Till Rasmussen, Mads Hvid Ribergaard, and Hoyeon Shi were founded by the Danish State through the National Centre for Climate Research. Axel Schweiger acknowledges funding from NASA Grant 80NSSC20K1253, ONR grants/contracts N00014-22-1-2346, N0002421D6400. Donghui Yi was funded under contract ST133017CQ0050/1332KP22FNEED0042.

Sang-Moo Lee (Seoul National University), Gorm Dybkjaer (Danish Meteorological Institute), and Suman Singha (Danish Meteorological Institute) contributed to the creation of the DMI_CS2-AMSR_AVHRR and DMI_CS2-IS2 datasets.



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
