# Peer review of "Biases, Uncertainties, and Trends in Arctic Sea-Ice Thickness: A Cross-Product Analysis from 1995 to 2023"

_EGUsphere, 2025_

## Referee Comment (RC1)

**Review of: Biases, Uncertainties, and Trends in Arctic Sea-Ice Thickness: A Cross-Product Analysis from 1995 to 2023.**

Robbie Mallett

I should say up front that I could be perceived to have a conflict here as I work in the same group as Jack Landy. But given the diversity of authors on this paper I think it will be hard to find totally detached reviewers. I therefore provide my review alongside a statement of certainty that it has been conducted impartially and without undue influence.

**1 Synopsis**

The manuscript is an analysis of a large database of sea ice thickness products produced by a variety of satellite instruments and models - the database is being published in parallel. The authors first evaluate the products against upward looking sonar observations from the Beaufort Sea, and investigate whether the stated product uncertainties correspond to the mismatch at the moorings (Figures 1 -3). They then conduct a pairwise intercomparison between the products in the database, before assessing temporal trends. Such an intercomparison effort is worthy, and follows in the sucessful footsteps of the IMBIE and GLAMBIE intercomparison projects. I therefore applaud ESA for funding the project, and the authors for collating a dataset which I agree has unprecedented comprehensiveness. However, I have a few concerns about this paper that lead to my recommendation of major revisions.

The first is that the method used to assess the skill of the products is dominated by the seasonal cycle of ice thickening at the BGEP moorings. This is an issue because many product users (including the authors when approaching their time-trend analysis) are interested in interannual variability and trend detection. The seasonal timescale that dominates the evaluation does not say much about this, so is of limited use to product users. The metric's narrow scope may be masking real development-driven improvements in product skill over time, and its ubiquity may be masking inter-product diversity via Goodhart's Law.

Secondly, given the presence of a companion data-description paper submitted to Scientific Data, I think this paper should be more comprehensive if it is to be published in TC. In terms of the inputs, it omits the sea ice thickness data produced by the UK Centre for Polar Observation and Modelling. This has historically been widely used, so I think will be of interest to many readers. But the paper also uses only a small fraction of hard-won in-situ and airborne SIT evaluation data. As a community we constantly request more evaluation data. On that basis we should at least use the data that are available in papers like this. The explanation that these data weren't analysed because nobody put them into the SIN'XS database surprised me, given that the authors were and presumably are still in charge of the database, and are also free to download and analyse additional data themselves in this manuscript. If this paper is not the right place for a full analysis, where is?

Finally, I'm curious about the Author Contributions statement, which suggests a major data curation effort was necessary for this paper, in addition to curation of the dataset already submitted to Scientific Data.

**2 Broad Comments**

**2.1 Seasonal Bias in Assessment Method**

I think the assessment method used in the paper (shown well in its Figure 1) leads to a flattering and quite narrow evaluation of sea ice thickness products. This is because when calculated in this way, the correlation and RMSD statistics are dominated by the fidelity with which a product captures the seasonal cycle of ice thickening rather than any interannual variability or trends. We studied this phenomenon in Nab et al. (2024), and concluded that the SIN'XS project should carefully consider its rubric for assessing a product's skill (and avoid the approach that has been taken here).

I won't recap the paper in full, but essentially the seasonal cycle of sea ice thickening is **much bigger** than interannual variability or decadal trends. So the top right of the manuscript's Figure 1a is occupied by climatologically thick sea ice in April, and the bottom left is occupied by climatologically thin sea ice in October, and the skill metrics are therefore sensitive to how well a product represents this climatology. If the top right of the plot were dominated by years where the sea ice were thicker than usual, then high scores would represent a good representation of interannual variability. If the top right were full of datapoints from one end of the timeseries, then high scores would represent good capture of temporal trends (to the extent that they exist). But since the plot is currently dominated by the seasonal cycle, the scores are dominated by the skill in this quite narrow regard.

The limitations that come from the dominance of the seasonal cycle in the assessment are neatly illustrated by construction and evaluation of a "broken" sea ice thickness product. I made this by imagining CryoSat-2/Sentinel-3 were simply returning radar freeboards of zero all year round. I converted these radar freeboards to sea ice thickness with the unmodified Warren snow climatology and compared all EASE grid pixels within 75 km of the BGEP moorings to match the authors' analysis. This hopefully represents the most trivial sea ice thickness product one can imagine, using no data actually measured by a satellite. It has no year-to-year variability or trend. My Figure 1 (below) suggests that it's very comparable in skill to the AWI, GSFC and CPOM products when $R^2$ and RMSE are calculated with the authors' method.

The reason the broken product performs well is because a seasonal climatology is encoded into it by the input of external snow data which reliably thickens over winter and thus reliably increases the derived thickness, even in the case where the radar data doesn't seasonally evolve at all.

Given the performance of my 'broken' sea ice thickness product in the manuscript's evaluation framework, I recommend further investigation of how the skill of a product on longer (perhaps more meaningful) timescales can be measured. Obviously I would say this having been part of Nab et al. (2024), but I do think one should consider that a product's skill is best assessed by its ability to capture **anomalies from its climatology**. When assessed in this way (i.e. with the seasonal cycle removed), my "totally broken" product rightly performs poorly, is meaningfully outperformed by the 'real' products. This methodology isn't new, having been used in Landy et al. (2022):

> '*Furthermore, after removing the climatological mean seasonal cycles of ice draft from the three long time series in the Beaufort Sea, the anomaly correlation coefficients between ULS and CryoSat-2 observations are 0.45, 0.51 and 0.37 for moorings A, B and D, respectively. This suggests that CryoSat-2 summer observations can capture a significant portion of the interannual variability in SIT recorded by moored ULS sensors.*'

But why should we care about this seasonal bias in the assessment method? Some product users will indeed value the faithful capture of the seasonal cycle, but many others want to know how much the sea ice in a region or on a route is thinner or thicker than usual (operational ice charters for instance). They therefore will want to know which products have higher or lower variability, and whether this variability is real and matches the "true" variability seen at the moorings. Other users (including the authors in this manuscript) want to assess temporal trends in thickness over a decade or more. The assessment structure in Figure 1 is inadequate for that because it's so heavily weighted towards capturing skill on seasonal timescales. To summarise this point: even if the products produce a convincing climatology, we can't trust the trends unless

[Figure]

Figure 1: Broken sea ice thickness product vs three other CryoSat-2 products. The broken one (top left - constructed with zero-radar freeboards and W99) has comparable $R^2$ and RMSE scores. The fact that it performs so well despite not showing no interannual variability or trend suggests that these metrics of intercomparison are of limited use when used in this way.

the products actually know whether the sea ice is thicker or thinner than average in a given year. But also the ubiquity of this fairly narrow assessment rubric may actually be masking real improvements (or deterioration) in IAV and trends that come out of product development.

I will finally add (and this is a minor point) that the ubiquity of this rubric in the literature raises the question of whether these metrics have become targets for product developers over time. This should incentivise other methods of intercomparison (and use of different evaluation data), because the statistics used here have already been quite extensively 'tuned to' by product developers. Analysing the seasonally-dominated statistics at BGEP moorings may therefore mask the true diversity between products because they're often tuned to score well in this particular analysis in this particular place.

**2.2 Comprehensiveness of evaluation data**

The concluding paragraph states:

> *'We would like to extend [the analysis] to other reference measurements, particularly airborne measurements (Icebird, Cryo-TEMPO, CryoVEx, OIB) and more ULS data (Lindsay, 2010) to account for a broader range of sea-ice types and regions'*

If the authors would like to do this, I'm curious about why it didn't happen? I can't think of a better team to do it; some authors on this paper are the leading experts in these in-situ and airborne observations (with the data often collected by them, or their own institutions). It would massively add value to the analysis, and without this analysis it's hard to interpret the results because by their own admission, the authors are limited by sea ice type and region. Noting that there are thirty authors on this paper and the distributed SIT products were pre-prepared and published anyway for another paper, this seems like a reasonable ask.

To give a concrete example, why not include the ULS moorings from the Fram Strait Arctic Outflow Observatory? These are publicly available at the appropriate time resolution, and the analysis method is basically identical to the BGEP moorings but on the other side of the Arctic. There are also similar moorings in the Laptev Sea that I believe authors on this paper will have access to (see Landy et al. (2022) for a previous evaluation of this type). On line 258 reads: *'Since no more ULS reference data are available in the SIN'XS database, we cannot extend our analysis to other regions'*. I would gently ask the authors: Why not just download these reference data onto your local machine? And were you not (at least partially) in charge of what went into the database?

The satellite sea ice thickness community often calls for more in-situ data for evaluation, and emphasizes their critical importance to funders. To be consistent with these calls, I think community assessments like this should use the full range of publicly available data or at least multiple sources. If the BGEP moorings are enough for us to properly evaluate our products, why do we spend so much money, carbon and time proposing and gathering more?

**3 Narrow/Technical Comments**

**3.1 Breaking out by sea ice type**

Going back to the concluding paragraph:

> *'We would like to extend [the analysis] to other reference measurements ... to account for a broader range of sea-ice types'*

If the authors strictly mean 'type' as in MYI and FYI, I think they can already split the analysis at the BGEP moorings into months when they're dominated by FYI cover vs MYI cover? See Figure 3 of Nab et al. (2024) for how this can be done. I totally agree that this is a desirable piece of analysis, particularly for some product users who are more sensitive to ice thickness biases over certain ice types. There's also clearly an argument that skill over FYI will become increasingly important for SIV estimates. But it would also be useful for product developers to identify routes to improvement, as well as for a discussion about why these products might be better or worse at the moorings in given years.

**3.2 Comprehensiveness of input SIT products**

The first sentence of the abstract reads:

> *Sea-ice thickness is a key component of the Arctic climate system, but yet a comprehensive assessment across observations and numerical models is still missing.*

There's a strong implication to the reader here that **this** is the missing comprehensive assessment. In the pitch to the editor for this becoming a highlight paper upon publication, the authors made this explicit:

> *We provide for the first time a reconciled estimate of Arctic sea-ice thickness which comprises the entire range of available products.*

Unless I've misunderstood, the CPOM sea ice thickness product described by Laxon et al. (2013) and Tilling et al. (2018) wasn't included? I think this limits any ability to say that the paper analyses 'the entire range of available products'. It's historically been one of the more popular sea ice thickness products and is publicly available and regularly updated. There are multiple products from some groups (UiT, LEGOS) - so why omit the CPOM product in what is implicitly (and to the editor and reviewers explicitly) described as a comprehensive assessment?

**3.3   What exactly has been curated here?**

There are thirty authors on this paper, which seems a lot for an article that investigates a dataset that was previously curated and submitted to a data journal as a separate publication. Normally I would regard this as an editorial concern rather than one for me as a reviewer, but I am curious that nine authors have defined their contribution to this manuscript solely as "Data Curation". What data curation was there to do, given the sea ice thickness data were already curated for the Scientific Data submission?

If the SIN'XS data submitted to Scientific Data required the dedication of nine additional authorships purely to subsequent "curation", then this casts some doubt on the usability of the SIN'XS data product, and also these *doubly-curated* data should be published and archived in their own right alongside this paper.

I'm on board with publication and detailed description of large or complex data in data journals, with subsequent analysis of the data in a more conventional journal such as TC. But those who self-describe as having solely curated the original data surely do not warrant authorship on a subsequent analysis paper to which they themselves declare they have not contributed analysis?

**3.4   Gridding vs Projecting**

I'm not an expert in this stuff, but I wonder whether L184 would be better phrased as 'regridded them to the EASE2 grid'? My loose understanding is that the EASE2 grids are specific sets of x/y coordinates where the coordinate space (in the NH) is a Lambert Azimuthal equal-area projection. So things are gridded to EASE2, but projected to Lambert Azimuthal? I appreciate this is confusing and sometimes even the NSIDC refers to the EASE "projections". If there's somebody on the authorship team (probably several people) that thinks 'Robbie is just wrong on this' then feel free to say that and ignore this comment.

**3.5   Regressions**

I'm not sure what the p-values correspond to in Figure 1. Presumably this is some kind of hypothesis test for a linear trend? The way it's presented, the reader might assume that the p-value somehow corresponds to the R-squared value, but to my knowledge R-squared statistics do not inherently come with a p-value. I suppose one could construct a hypothesis test that given a hypothetical random distribution, and then calculate the p-value for an R2 statistic that's at least as extreme as the value produced. But that seems unnecessary here since there's obviously a relationship and the interesting science surrounds the strength of the relationship rather than its statistical significance as indicated by a p-value.

**3.6   Dual-frequency stuff**

> '*It also highlights the necessity of dual-frequency KuKa measurements (Willatt et al., 2025), which can be used to derive snow thickness directly*'

In that paper we saw that almost no co-polarised power came from the ice surface, so insofar as the KuKa instrument we used simulates CRISTAL (and it may well not do because of issues of length-scale and beam-geometry), the results imply that CRISTAL would not have 'worked' on that particular snow. So if anything, that paper goes in the pile that casts doubt on the CRISTAL dual-frequency method (although such a classification is obviously superficial, and surface-to-satellite comparisons are hard as I've mentioned). There are many papers in the pile that supports the dual-frequency operating principle of CRISTAL (e.g. Landy et al., 2026), but I wonder if citing Kern et al. (2020) may actually be best here for a reader who's not familiar with upcoming missions.

**3.7  Snow Density**

'*a daily parametrised snow density based on (Mallett et al., 2020)*'

I didn't intend for that equation to be used so widely, and there are unfortunately quite a few downsides to what I did. I recently published an easier to use 'updated' version of this equation that is similar in form but more methodologically robust (Mallett, 2025). Suggest that would be better.

**3.8  Other things**

'*the observed pause in sea-ice extent decline in the last twenty years is statistically robust*'

This statement is quite vague. I feel like it's a statement about whether it is statistically significant by comparison to the overlay of internal variability on the forced response? If so, it should be more specific.

Should there be units of [m] on RMSD in Figure 6, and Bias in Figure 4, and the heatmap of Figure 3?

L49: I'm not sure altimeters do cover the full range of SIT - they really struggle when it's below 20 cm or so, hence the mergers with SMOS etc

Related to the above, why do none of the heatmaps contain BGEP measurements of less than 0.2m? My analysis of the BGEP mooring data (see figure above) definitely does produce mean thicknesses for some months in this 0 - 0.2 m range.

Furthermore on Figure 1, I'm suspicious that the "core" of the data cloud on panel (a) is blown out, i.e. it's passed the max of the colorbar. This is perhaps happening on panel c too, but to a lesser extent. Normally this phenomenon is indicated by a small arrow on top of the colorbar (extend keyword in matplotlib), but more usefully one could just normalise the colormaps so that colors represent *% of datapoints* - this would allow better representation of the range I think.

**References**

Kern, M., Cullen, R., Berruti, B., Bouffard, J., Casal, T., Drinkwater, M. R., Gabriele, A., Lecuyot, A., Ludwig, M., Midthassel, R., Navas Traver, I., Parrinello, T., Ressler, G., Andersson, E., Martin-Puig, C., Andersen, O., Bartsch, A., Farrell, S., Fleury, S., Gascoin, S., Guillot, A., Humbert, A., Rinne, E., Shepherd, A., van den Broeke, M. R., and Yackel, J.: The Copernicus Polar Ice and Snow Topography Altimeter (CRISTAL) high-priority candidate mission, The Cryosphere, 14, 2235–2251, https://doi.org/10.5194/tc-14-2235-2020, 2020.

Landy, J. C., Dawson, G. J., Tsamados, M., Bushuk, M., Stroeve, J. C., Howell, S. E. L., Krumpen, T., Babb, D. G., Komarov, A. S., Heorton, H. D. B. S., Belter, H. J., and Aksenov, Y.: A year-round satellite sea-ice thickness record from CryoSat-2, Nature 2022 609:7927, 609, 517–522, https://doi.org/10.1038/s41586-022-05058-5, 2022.

Landy, J. C., de Rijke-Thomas, C., Nab, C., Lawrence, I., Glissenaar, I. A., Mallett, R. D. C., Fredens-borg Hansen, R. M., Petty, A., Tsamados, M., Macfarlane, A. R., and Braakmann-Folgmann, A.: Antici-pating CRISTAL: an exploration of multi-frequency satellite altimeter snow depth estimates over Arctic sea ice, 2018–2023, The Cryosphere, 20, 183–208, https://doi.org/10.5194/TC-20-183-2026, 2026.

Laxon, S., Giles, K. A., Ridout, A. L., Wingham, D. J., Willatt, R., Cullen, R., Kwok, R., Schweiger, A., Zhang, J., Haas, C., Hendricks, S., Krishfield, R., Kurtz, N., Farrell, S., and Davidson, M.: CryoSat-2 estimates of Arctic sea ice thickness and volume, Geophysical Research Letters, 40, 732–737, https://doi.org/10.1002/grl.50193, 2013.

Mallett, R. D.: A methodologically robust densification function for snow on multiyear Arctic sea ice, Journal of Glaciology, 71, e24, https://doi.org/10.1017/JOG.2025.5, 2025.

Mallett, R. D. C., Lawrence, I. R., Stroeve, J. C., Landy, J. C., and Tsamados, M.: Brief communication: Conventional assumptions involving the speed of radar waves in snow introduce systematic underesti-mates to sea ice thickness and seasonal growth rate estimates, Cryosphere, 14, 251–260, https://doi.org/10.5194/tc-14-251-2020, 2020.

Nab, C., Mallett, R., Nelson, C., Stroeve, J., and Tsamados, M.: Optimising Interannual Sea Ice Thickness Variability Retrieved From CryoSat-2, Geophysical Research Letters, 51, e2024GL111 071, https://doi.org/10.1029/2024GL111071, 2024.

Tilling, R. L., Ridout, A., and Shepherd, A.: Estimating Arctic sea ice thickness and volume using CryoSat-2 radar altimeter data, Advances in Space Research, 62, 1203–1225, https://doi.org/10.1016/j.asr.2017.10.051, 2018.

Willatt, R., Mallett, R., Stroeve, J., Wilkinson, J., Nandan, V., and Newman, T.: Ku- and Ka-Band Polari-metric Radar Waveforms and Snow Depth Estimation Over Multi-Year Antarctic Sea Ice in the Weddell Sea, Geophysical Research Letters, 52, e2024GL112 870, https://doi.org/10.1029/2024GL112870;PAGE:STRING:ARTICLE/CHAPTER, 2025.